# White matter disconnectivity fingerprints causally linked to dissociated forms of alexia

Sam Ng [1,2 ✉], Sylvie Moritz-Gasser[1,2,3], Anne-Laure Lemaitre[1,2], Hugues Duffau [1,2] &
Guillaume Herbet [1,2,3]

For over 150 years, the study of patients with acquired alexia has fueled research aimed at disentangling the neural system critical for reading. An unreached goal, however, relates to the determination of the fiber pathways that root the different visual and linguistic processes needed for accurate word reading. In a unique series of neurosurgical patients with a tumor close to the visual word form area, we combine direct electrostimulation and population-based streamline tractography to map the disconnectivity fingerprints characterizing dissociated forms of alexia. Comprehensive analyses of disconnectivity matrices establish similarities and dissimilarities in the disconnection patterns associated with pure, phonological and lexical-semantic alexia. While disconnections of the inferior longitudinal and posterior arcuate fasciculi are common to all alexia subtypes, disconnections of the long arcuate and vertical occipital fasciculi are specific to phonological and pure alexia, respectively. These findings provide a strong anatomical background for cognitive and neurocomputational models of reading.

[1] Department of Neurosurgery, Gui de Chauliac Hospital, Montpellier University Medical Center, Montpellier, France. [2] Institut de Génomique Fonctionnelle, Université de Montpellier, CNRS, INSERM, Montpellier, France. [3] Department of Speech-Language Pathology, University of Montpellier, Montpellier, France. ✉email: s-ng@chu-montpellier.fr

Early descriptions of brain-damaged patients with isolated acquired reading disorders[1] (i.e. alexia) have raised a fascinating paradox: how can brain areas be specifically devoted to reading despite the fact that lexicality is, from a phylogenetic standpoint, a recent cultural invention (~5000 years old)? While it seems unlikely that the brain holds in-built neural circuits dedicated to reading at birth, it has now been established that learning to read partly hinges on the progressive specialization of well-situated areas interfacing the visual to the language system. In particular, convergent findings indicate that the brain follows a dynamic remodeling during reading acquisition, including an intense refining of the entire brain paths dedicated to language[2,3]. Thanks to the use of functional MRI (fMRI), the network of cortical areas modulated by, and possibly critical for efficient reading is increasingly understood[4–6]. There is, however, a great deal of uncertainty about which neural connections might integrate information within the reading network.

One of the major advances in the field was the description of the visual word form area (VWFA)[7], a circumscribed area lodged in the mid-portion of the left fusiform gyrus (generally within the occipitotemporal sulcus). Although the specificity of this area in decoding written language is still debated[8], the VWFA is firmly established as a crucial and reproducible site for visual orthographic representations[9–11] and to act as a pivotal interfacing area for reading, by allowing the ventral visual system and the perisylvian language network to functionally interact. Injury to the VWFA typically causes what has been called pure alexia (Pu-A), i.e. a severe inability of recognizing words, and sometimes letters, with roughly spared spoken language and vision. In current feed-forward models of reading, it is suggested that, after the word has been recognized and encoded within the VWFA, two concurrent but interactive language networks are solicited. First, a dorsal phonological route allows the transcoding of letter clusters (graphemes) to appropriate speech sounds (phonemes). As this system is critical for sublexical units to be "assembled" phonetically, disruption to this pathway leads to inaccurate reading of "pseudowords" (e.g. heth, vun or sark in English), namely phonological alexia (Ph-A). In parallel, a ventral lexical-semantic route allows holistic access to the sound of a whole word from a mental store of word forms[12] and meanings[13]. In alphabetic writing systems, an impairment of the lexical-semantic route results in difficulties in reading "irregular" words that violate typical letter-sound correspondences (e.g. friend or yacht in English) with an over-reliance on classical sublexical assembly procedure, leading to "regularization" errors. Such difficulties are referred to as surface alexia or lexical-semantic alexia (Ls-A).

Besides, attempts to identify critical lesion sites in Ph-A have yielded inconsistent results throughout the perisylvian structures[14–16]. Anatomical correlates to the lexical-semantic route are even more debated, especially regarding the divergent participation of the anterior temporal lobe and the posterior temporal areas[5,16–18]. Overall, fMRI-based studies have evidenced neuroanatomical dissociations between phonological and lexical-semantic routes, with distinct patterns of activation within the left supramarginalis, posterior middle temporal and fusiform gyri in phonological processing[19], and the left lateral anterior temporal lobe[5], middle temporal gyrus, angular gyrus, precuneus and posterior cingulate in lexical-semantic processing[19]. However, few modern lesion studies have established the differential causal contributions of specific brain regions to specific reading impairments[20,21]. Importantly, most lesion studies are based on patients presenting with stroke injury, allowing a very limited lesion coverage of the middle temporal gyrus, the inferior temporal gyrus and the VWFA. This is also true for the fiber pathways emanating or interconnecting these brain areas.

Although cerebral disconnection was historically suggested as a possible pathophysiological mechanism of alexia[1], only limited pieces of evidence have been provided regarding the white matter connections that may be concerned with propagating neural signals within the reading network. Studies evaluating the impact of literacy on structural connectivity using diffusion tensor imaging have suggested a possible role of the posterior arcuate fasciculus (AF), as its microstructural properties greatly covary with the development of reading expertise[3]. On the other hand, a small handful of either neuropsychological[22] or neuromodulation[23] works have provided experimental supports for a role of the left ILF in reading, as its transient or definitive disruption is associated with Pu-A. In view of the connectivity patterns of the VWFA, other tracts may also contribute to reading including the inferior fronto-occipital fascicle (IFOF)[24] and the vertical occipital fascicle (VOF)[10,24], since both have projections to the VWFA. However, there are no neuropsychological studies providing clear-cut anatomo-functional correlations between these tracts and the existing forms of alexia, with the notable exception of the posterior part of the ILF[22,23].

Direct electrostimulation (DES) mapping in patients undergoing awake low-grade glioma removal with online neurocognitive monitoring is currently the gold standard method to allow optimal resection and preservation of neurological[25,26] and cognitive functions[27]. The neuroscientific advantage of electrostimulation over other investigation approaches lies in its ability to directly investigate, with an excellent spatiotemporal resolution, the role of white matter connectivity in cognition and behavior and to capture the interindividual variations in the neural representation of functions[28–31]. It is thus an effective means of identifying which cerebral disconnections may be critical for a given neuropsychological impairment. Its value is furthermore heightened when combined with tractographic methods that allow to delineate the neural connections impaired by the stimulation trains[32].

In this study, we aimed to clarify the disconnectivity patterns associated with the more common forms of alexia (i.e. phonological, lexical-semantic and pure alexia). In particular, we were interested in establishing whether each type of alexia could be distinguished by a particular white matter disconnectivity fingerprint. To do this, we combined in vivo axonal DES in a rare series of patients with a low-grade glioma diffusing in the vicinity of the VWFA and population-based, brainwide connectome analyses.

## Results

**Effects of direct electrostimulation on reading abilities.** Biphasic axonal DES (60 Hz, 1 ms pulse width, mean intensity of $2.52 \pm 0.58$ mA ranging from 1.75 to 3.50 mA) caused transient difficulty in reading words in 25 patients. A reading impairment was defined as an inability to read correctly a single word within a 4-s time frame. Intraoperative findings were recorded and interpreted by a senior speech-language pathologist who remained blinded to DES application. Overall, 72 white matter sites were identified (27 eliciting reading disturbances and 45 eliciting language disturbances). Note that two distinct sites of alexia were identified in two patients. A complete inability to read all word categories without additional verbal language impairments (i.e. the typical pattern for pure alexia; hereafter Pu-A) was identified in seven patients. In four patients, axonal DES caused difficulty in reading pseudowords with a preserved ability to read both regular words and irregular words (i.e. a phonological alexia; hereafter, Ph-A). Lastly, axonal DES induced misreading of irregular words (i.e. lexical-semantic alexia; hereafter Ls-A) in 16 patients.

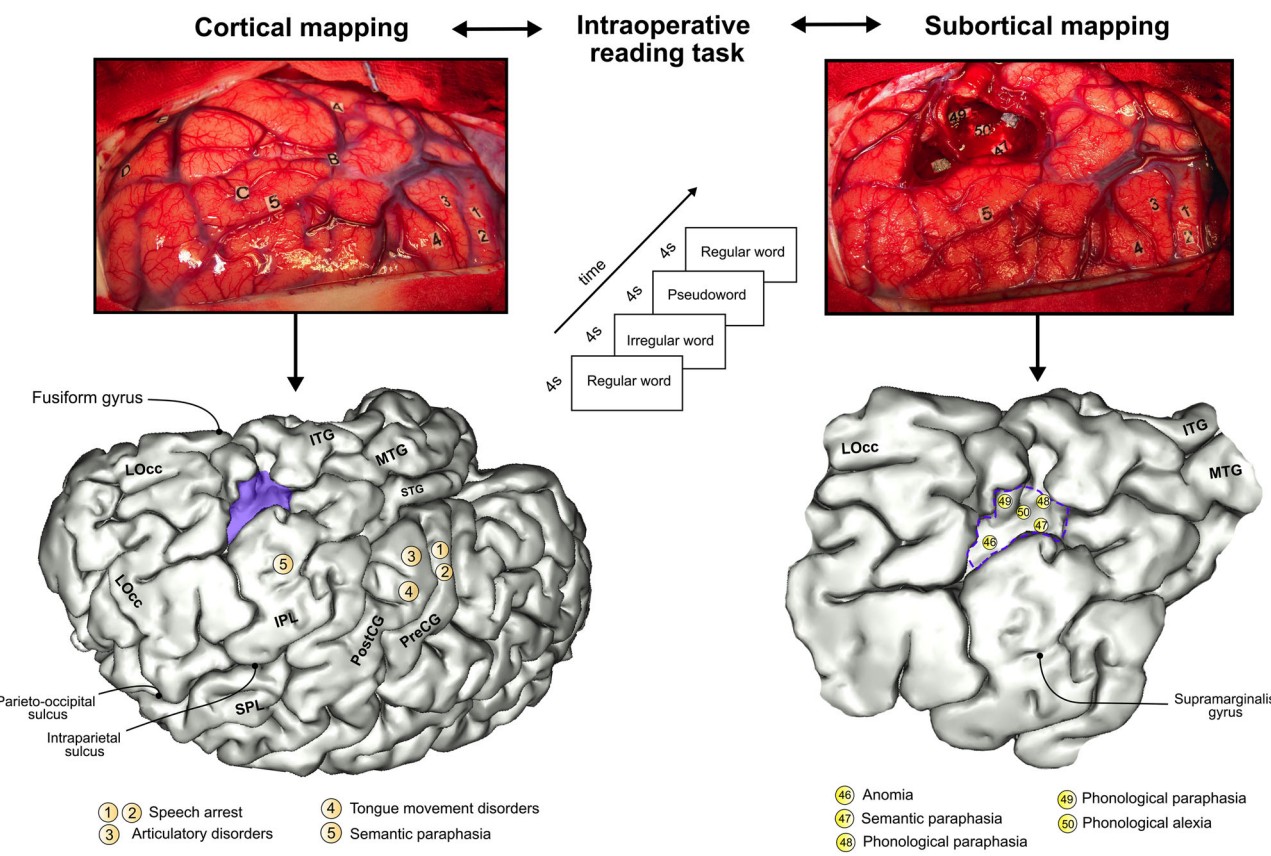

**Fig. 1 Illustration of a cortical and subcortical mapping.** DES applied during cortical mapping prior to surgical resection (upper left photograph). Tag letters indicate the boundaries of the tumor, identified with intraoperative echography. Intraoperative protocol for reading tasks (central scheme). DES applied during axonal mapping allowed the detection of functional boundaries, driving the surgeon to stop the resection (upper right photograph). The bottom part of the figure illustrates a 3D template reporting the stimulation sites obtained in this illustrative case. Montreal Neurological Institute coordinates were obtained by direct surface-matching of the 3D template, using BrainVISA/Anatomist 5.0 software. ITG inferior temporal gyrus, LOcc lateral occipital cortex, MTG middle temporal gyrus, PostCG postcentral gyrus, PreCG precentral gyrus, SPL superior parietal lobule, STG superior temporal gyrus.

An overview of both the intraoperative procedure and the method used to accurately position DES sites associated with functional responses is displayed in Fig. 1. To appreciate the spatial topography of tumor resections, an overlap map is provided in Fig. 2; the maximum density occurred in the left inferior temporal gyrus, in front of the fusiform gyrus and the occipital-temporal sulcus (i.e. the typical location of the VWFA). The complete distribution of sites related to reading and verbal language are available in Supplementary Fig. 1. Montreal Neurological Institute (MNI) coordinates are provided in Supplementary Tables 1 and 2.

**Disconnectivity fingerprints of alexia subtypes.** Here we were especially interested in identifying the white matter disconnectivity fingerprint of each alexia subtype. To this end, 2.5-mm (in radius) spherical VOIs (representing each responsive white matter site) were individually positioned in the 152-MNI space and embedded into the HCP-1021-1mm dataset. Fibers intersecting with each VOI (i.e. the "disconnected" fibers) were reconstructed (see "Materials and Methods" for details regarding the parameters used for the tracking). Next, pairwise white matter disconnectivity matrices using cortical parcels from the Brainnetome atlas[33] were generated. The resulting left-hemispheric disconnectograms are displayed in Fig. 3 and revealed dissimilarities across alexia subtypes.

The disconnectivity pattern of Ph-A was characterized by fronto-temporal (involving the inferior frontal gyrus [IFG] and the middle frontal gyrus [MFG] on the one hand, and the middle temporal gyrus [MTG] and the inferior temporal gyrus [ITG] on the other hand) and parieto-temporal disconnections (involving pairwise disconnections between the inferior parietal lobule [IPL] and both the MTG and ITG) (Fig. 3a). The gyrus supramarginalis (referred as IPL_L_6_4) accounted for 91.3% of disconnections within the IPL, mainly related to paired disconnections with the ITG and MTG, which were very specific to Ph-A. Disconnections of the frontal lobe including the dorsolateral prefrontal cortex (MFG_L_7_2), the ventral premotor cortex (IFG_L_6_1 and 6_6) and the precentral gyrus (PrG_L) respectively accounted for 27.9%, 38.0%, and 14.0% of the frontal disconnections. The anterior-lateral part of the VWFA (ITG_L_7_7) accounted for 34.5% of temporal disconnections.

The disconnectivity fingerprints of Ls-A was clearly different. It was characterized by temporo-occipital, fronto-occipital and parieto-occipital disconnections (Fig. 3b). Most of the frontal lobe disconnections were related to the orbital gyrus (OrG_L_6_4). Paired disconnections between the IPL and the MTG/ITG/Fusiform gyrus involved noticeably the angular gyrus (IPL_L_6_5 and IPL_L_6_1). Interestingly, no disconnections within the gyrus supramarginalis were found. The posterior-lateral part of the VWFA (ITG_L_7_1) accounted for 31.5% of temporal disconnections.

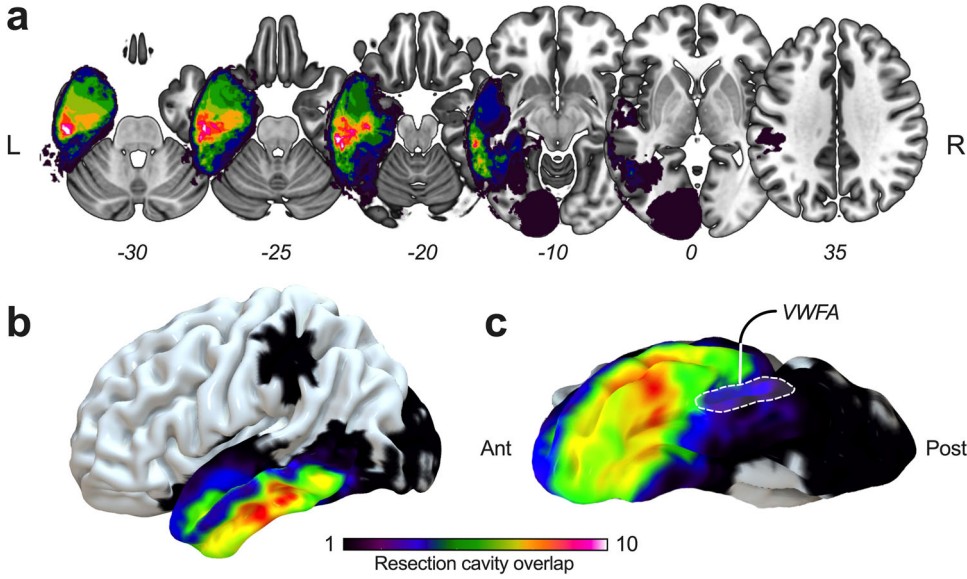

**Fig. 2 Spatial topography of surgical resections. a** Multiple axial sections, **b** three-dimensional lateral view, and **c** three-dimensional inferior view showing cortical projections of the resection cavity overlap. The maximum frequency of overlap ($n = 11$) was reached anteriorly to the visual word form area.

Lastly, Pu-A was mainly associated with temporal/fusiform-occipital and intra-occipital disconnections and to a much less extent with temporo-parietal or parieto-occipital disconnections (Fig. 3c). The mesial part of the VWFA (FuG_3_3) accounted for 39.4% of temporal disconnections, whereas paired intra-occipital disconnections accounted for 40.3% of all occipital disconnections.

In an attempt to further identify which pairwise disconnections were the most associated with each alexia subtype, we conducted pairwise comparisons between disconnectivity matrices related to Ph-A, Ls-A and Pu-A using two-tailed non-parametric Mann–Whitney tests. These analyses resulted in $n$ 123 × 123 matrices in which each matrix coordinate held a $Z$-statistics. These matrices were further plotted in the form of heatmaps (Fig. 4a–c). Disconnections that were considered different from one alexia subtype to another (i.e. $Z > 1.96$, $p < 0.05$ uncorrected) are displayed in Fig. 4a–c. Note that the following comparisons have to be interpreted with caution, since no adjustments for multiple comparisons were performed due to the limited amount of data for each alexia subtype, especially for phonological alexia ($n = 4$) and, conjointly, the considerable number of statistical comparisons. Overall, pairwise connections between MFG and ITG ($Z$ ranged from 2.37 to 3.02; uncorrected $p$ value ranged from 0.02 to 0.002) and between IFG and ITG ($Z$ ranged from 1.98 to 2.65; uncorrected $p$ value ranged from 0.04 to 0.008) were more frequently disrupted in the event of Ph-A compared with Ls-A and Pu-A. Interestingly, impaired connections between the IFG and the anterior-lateral part of the VWFA (ITG_L_7_7) and between the MFG and the anterior-lateral part of the VWFA were specifically found in the event of Ph-A. In addition, impaired ITG-angular connections had a higher propensity to be disrupted in Ls-A than in Pu-A ($Z = 1.97$; $p = 0.04$ uncorrected).

**Tract-level analyses**. To better delineate the patterns of disconnectivity described above, the total number of generated fibers using 2.5-mm spherical VOIs were projected onto the MNI space along with the HCP population-averaged white matter atlas (i.e. the HCP 1065 white matter atlas)[34]. We filtered the white matter fibers crossing the VOIs and belonging to the main associative white matter tracts. It is important to note that the nomenclature varies regarding the AF system. In the used atlas, the frontal-to-temporal portion, equivalent to the long segment of the AF[35] is referred as the AF per se and the parietal-to-temporal portion, equivalent to the posterior segment of the AF[35,36] is referred as the parietal-aslant tract. To maintain consistency with previous literature about reading, we used as terminologies "long segment" (lsAF) and "posterior" segment (pAF) of the AF. The number of "disconnected" tracts (by alexia subtype) within the lsAF, the pAF, the inferior longitudinal fasciculus (ILF), the IFOF and the VOF are provided in Fig. 5a. Overall, the ILF and the pAF were impaired in all types of alexia without significant differences between Ph-A, Ls-A and Pu-A. Statistical comparisons revealed that fibers of the lsAF were more frequently impaired by stimulation in the event of Ph-A compared with Ls-A ($p_{corrected} < 0.0005$) and with Pu-A ($p_{corrected} < 0.0003$). Likewise, fibers belonging to the VOF were more frequently stimulated in the event of Pu-A compared with Ls-A ($p_{corrected} < 0.01$). Language impairments were used as control stimulations. There were no statistical differences in the number of disconnected fibers between stimulations eliciting Ph-A and phonological paraphasias (Supplementary Fig. 2). The VOF was more frequently stimulated in Pu-A compared to phonological paraphasia ($p_{corrected} = 0.04$), semantic paraphasia ($p_{corrected} = 0.003$) and anomia ($p_{corrected} = 0.001$). Other results indicated that the lsAF was more frequently stimulated in case of phonological paraphasia compared to Ls-A and Pu-A ($p_{corrected} < 0.0001$ and 0.0008, respectively) and in the event of Ph-A compared to semantic paraphasia and anomia ($p_{corrected} = 0.01$ and 0.03, respectively). To control for the impact of using variable (but close) stimulation intensities, we repeated the same analyses by using VOIs with greater radius (i.e. 3 and 3.5 mm). The results of statistical analyses comparing Ph-A vs Ls-A vs Pu-A were unchanged (see Supplementary Fig. 2 for more details).

To provide more comprehensive analyses, streamline tractography maps generated for each stimulation site were then converted into 3D binarized maps. Between-group, voxelwise comparisons were then performed at the brainwide level to identify voxels that were more frequently and specifically "disconnected" for each kind of alexia. The subsequent FDR-corrected statistical maps indicated that fibers of the AF were more impaired for Ph-A compared to Ls-A and Pu-A (Fig. 5b).

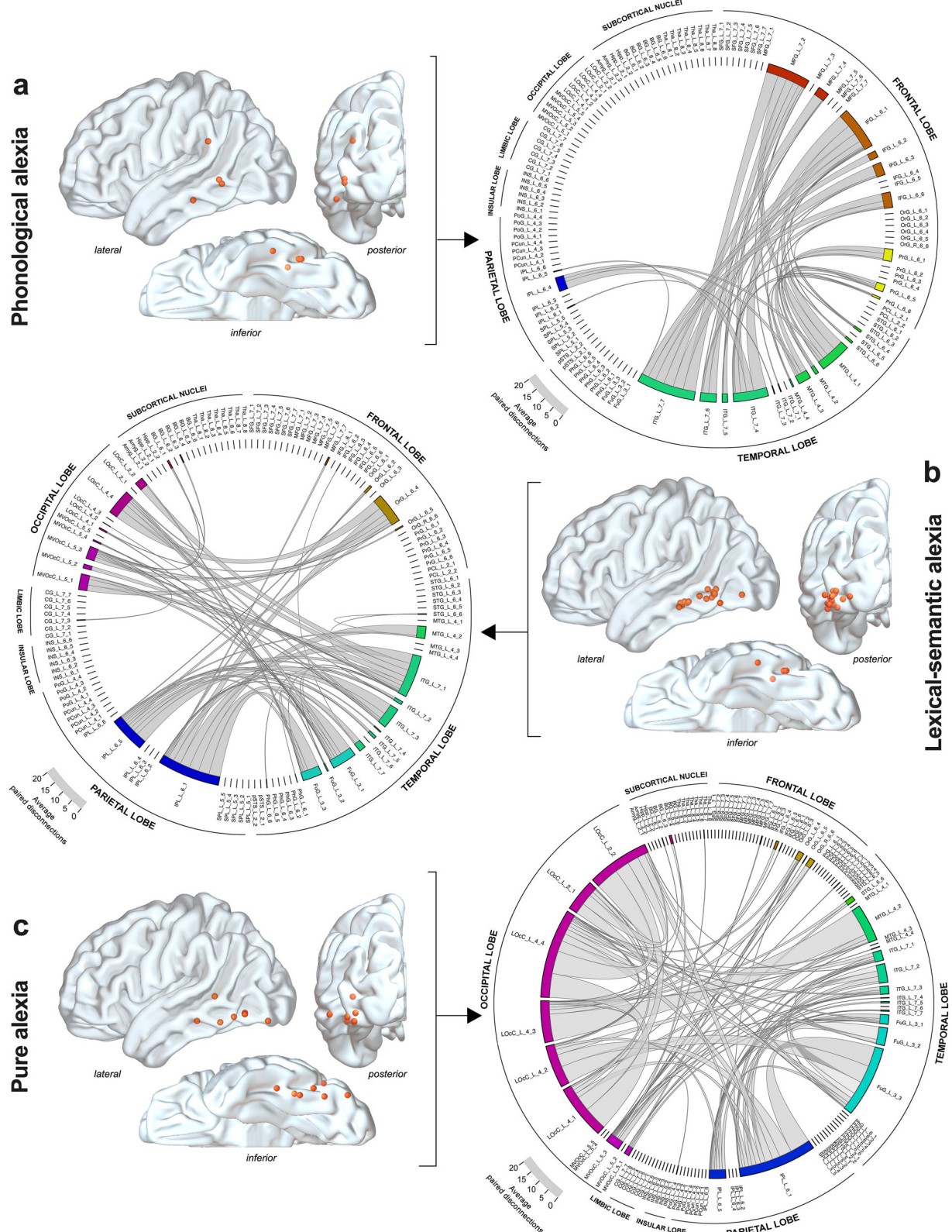

## Discussion

Cerebral disconnections are a common pathophysiological mechanism across multiple brain conditions[37–39], and probing their neuropsychological consequences is of vital interest for an advanced understanding of human functional networks. Using a method capable of producing well targeted but transitory disconnective breakdown, we searched for the neural connections causing dissociated forms of alexia when transiently inactivated. We further embedded the unmasked critical white matter loci in the HCP population-averaged connectome to generate the disconnectivity fingerprint of each alexia subtype. Overall, we found dissociated disconnectivity patterns which are discussed in the following.

Acquired deficits of reading pseudowords (i.e. Ph-A) has been reportedly associated with diffuse lesions targeting the perisylvian

**Fig. 3 Disconnectograms by alexia syndrome within the left hemisphere.** Axonal DES sites eliciting **a** phonological alexia ($n = 4$), **b** lexical-semantic alexia ($n = 16$), and **c** pure alexia ($n = 7$) are illustrated as orange spherical volumes of interest. Disconnectograms are displayed as chord diagrams, based on the parcellation from the Brainnetome atlas. Each connection inside the diagram illustrates a number of pairwise disconnection. The thickness of the chords is proportional to the average number of pairwise disconnection between brain areas. The location of the labels along the perimeter of the diagram is shifted from one diagram to another, depending on the average number of disconnections in a given diagram. Amyg amygdala, BG basal ganglia, CG cingulate gyrus, FuG fusiform gyrus, IFG inferior frontal gyrus, INS insula, IPL inferior parietal lobule, ITG inferior temporal gyrus, LOcC lateral occipital cortex, MFG middle frontal gyrus, MTG middle temporal gyrus, MVOcC medioventral occipital cortex, OrG orbital gyrus, PCL paracentral lobule, Pcun precuneus, PgF parahippocampal gyrus, PoG postcentral gyrus, PrG precentral gyrus, pSTS posterior superior temporal sulcus, SFG superior frontal gyrus, SPL superior parietal lobule, STG superior temporal gyrus, Tha thalamus.

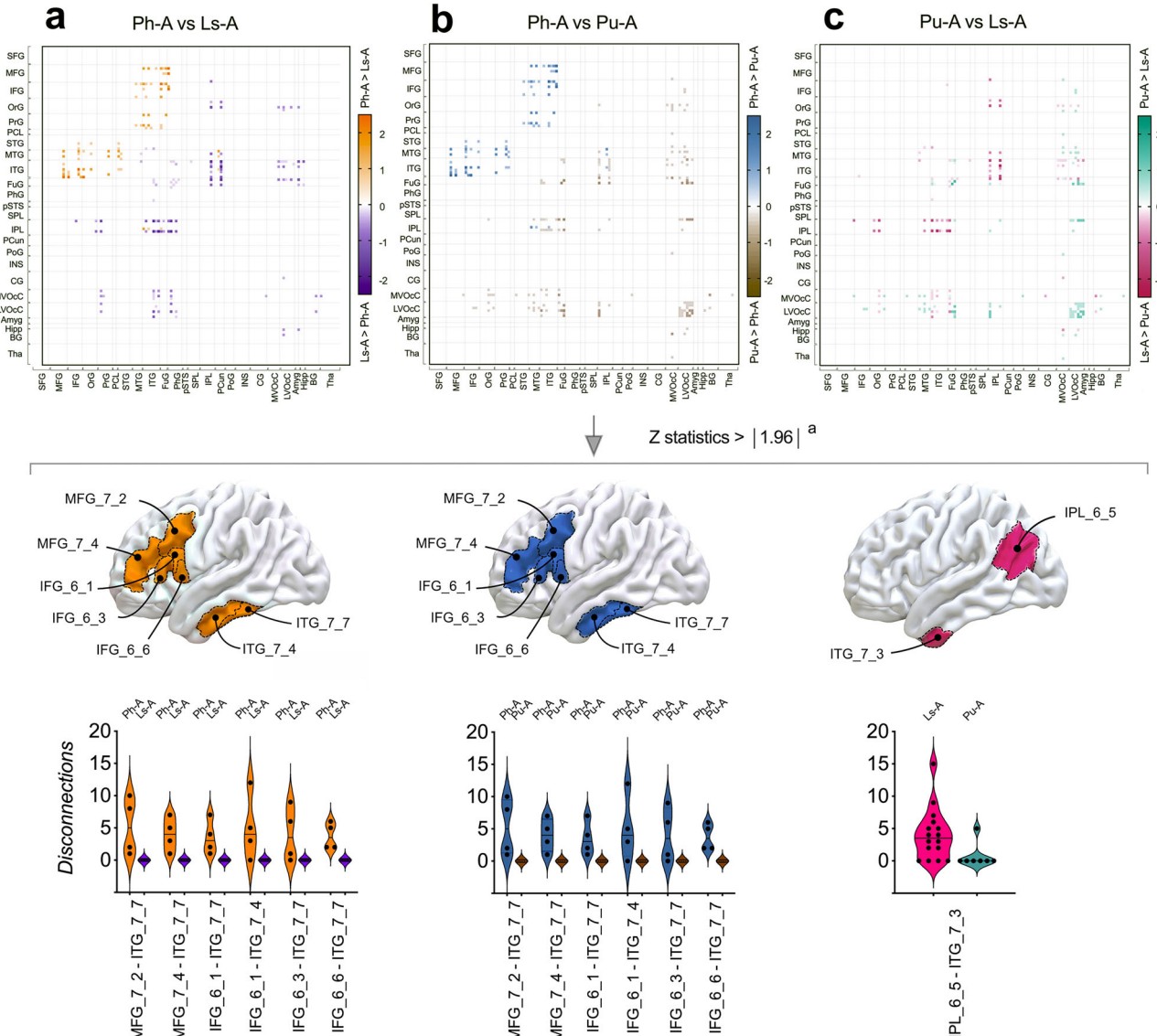

**Fig. 4 Comparisons of disconnectivity fingerprints.** Comparisons between pairwise disconnections in **a** phonological alexia (Ph-A) and lexical-semantic alexia (Ls-A), **b** Ph-A and pure alexia (Pu-A), **c** Ls-A and Pu-A, for each pairwise combination from the Brainnetome atlas. Heatmaps are presented in the upper panel. Anatomical parcels showing $Z$-statistics > |1.96| are illustrated in the middle panel. Higher rate of disconnections in Ph-A in comparison to Ls-A are illustrated in orange. Higher rate of disconnections in Ph-A in comparison to Pu-A are illustrated in blue. Higher rate of disconnections in Ls-A in comparison to Pu-A are illustrated in pink. Pairwise disconnections with $Z$-statistics > |1.96| are illustrated as violin plots in the lower panel. Individual values are presented as dots. Horizontal lines represent the median. Serial two-tailed Mann–Whitney comparisons were performed. IFG inferior frontal gyrus, IPL inferior parietal lobule, ITG inferior temporal gyrus, MFG middle frontal gyrus. [a]No corrections for multiple comparisons were applied.

areas[14,40]. Importantly, our findings are consistent with previous lesion mapping and fMRI studies suggesting that phonological decoding relies on perisylvian circuits[41]. Connections projecting into the gyrus supramarginalis via the pAF were associated with

Ph-A, although such connectivity profile was not confirmed by further pairwise disconnection comparisons. This result is concurring with the established contributions of the gyrus supramarginalis in reading accurately regular words[21], in reading aloud

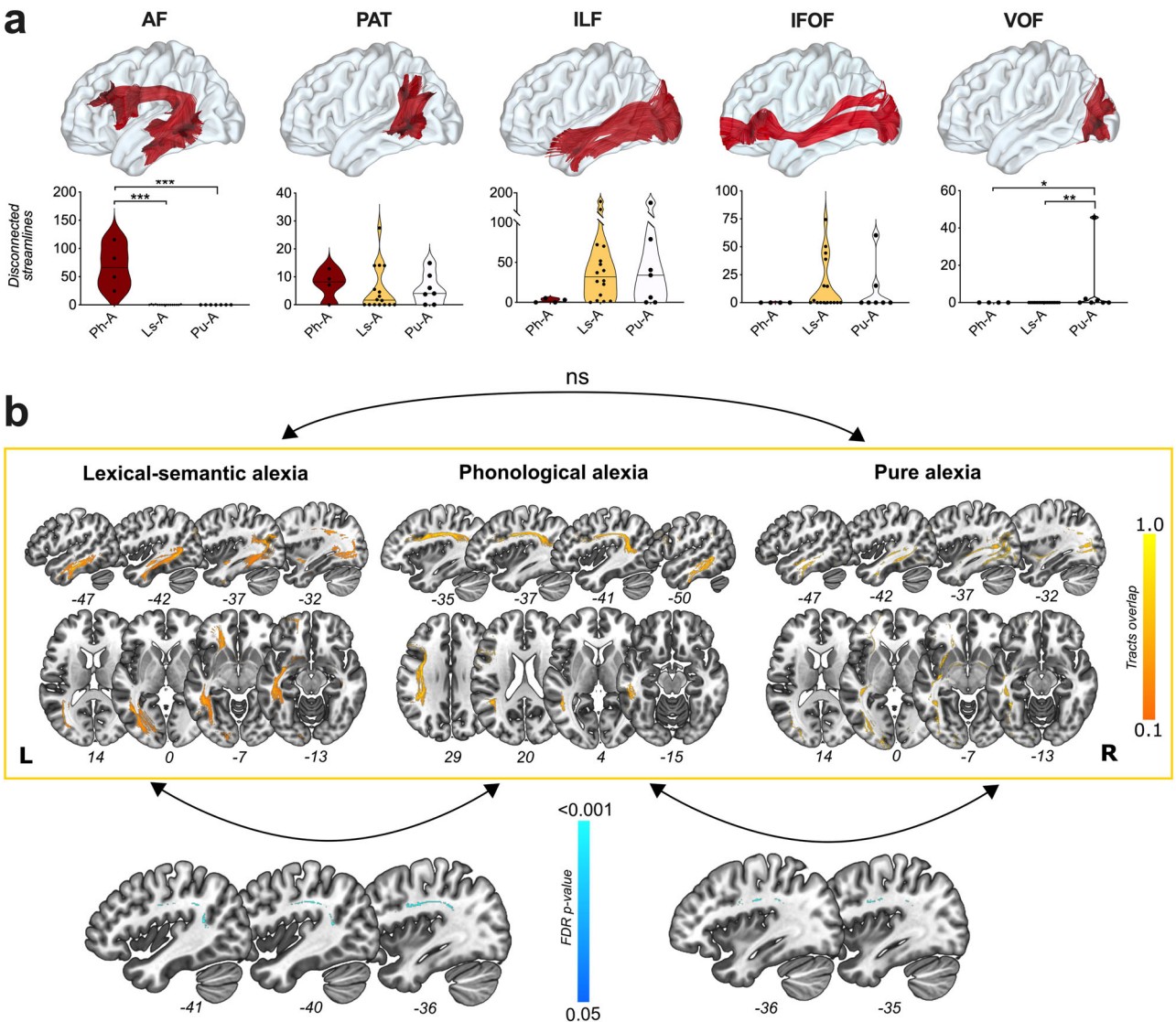

**Fig. 5 Tract-level analyses. a** Main white matter tracts (lsAF and pAF long segment and posterior of arcuate fasciculus, PAT parietal-aslant tract, ILF inferior longitudinal fasciculus, IFOF inferior fronto-occipital fasciculus, VOF vertical occipital fasciculus) according to the HCP 1065 white matter tractography atlas and number of disconnected fibers in each form of alexia (Ph-A phonological alexia, Ls-A lexical-semantic alexia, Pu-A pure alexia). Individual values are presented as dots. Horizontal lines represent the median. **b** Overlap of tracts being disconnected in each form of alexia. Voxel-based comparison between Ls-A vs Pu-A does not reach statistical significance. Voxel-based comparisons between Ls-A vs Ph-A (bottom left corner of the figure) and between Ph-A vs Pu-A (bottom right corner of the figure) highlight significant differences within the arcuate fasciculus in both comparisons. Ns non-significant.

compared to silent reading[42], in phonological retrieval during object naming[43] and in pseudowords recall[44]. Impaired connections between the frontal (mainly including IFG, MFG and the premotor cortex) and the inferior temporal cortex were also highly characteristic of Ph-A. Interestingly, it is known that the inferior lateral prefrontal cortex together with the adjacent premotor areas are critically involved in phonological processing[40,45], which is fully in line with the fact that we isolated lsAF as a critical pathway associated with the phonological route of reading. Such a finding endorses previous functional neuroimaging reports showing an increased functional connectivity between the precentral gyrus and the ventral occipital-temporal regions during reading of regular words[5]. Taken as a whole, our results suggest that the dorsal stream of language (mainly including the lsAF and the pAF) may mediate the ability to read pseudowords, as its disconnective breakdown causes Ph-A.

The lexical-semantic route is currently the less understood of the two-hypothesized linguistic-related reading pathways. Early descriptions of Ls-A in patients with semantic dementia established the concomitance of semantic impairment and difficulty in reading irregular words following progressive degeneration of the anterior temporal lobe[46]. Since then, the contribution of the semantic system in reading aloud has been a matter of debate. Recent findings have highlighted the role of the anterior temporal lobe as an important component of the ventral semantic system[5,47], while other studies have suggested that lexical-semantic retrieval might rely on other critical regions such as the anterior IFG and the orbitofrontal gyrus (areas that are critical for semantic control)[47], especially during concrete word reading tasks[21]. From a structural connectivity perspective, our results provide evidence that Ls-A is caused by the disruption of white matter fibers projecting in the anterior temporal lobe and the

orbitofrontal gyrus, and belonging to the ILF and IFOF, respectively. However, it remains unclear whether or not all portions of the ILF contribute equally to reading processing, since stimulations targeting the anterior ILF (namely, the segment of the ILF anterior to the VWFA) were found to rarely induce alexia[23]. In addition, we observed that Ls-A was also associated with disconnection of fibers interconnecting the angular gyrus and the ITG or the fusiform gyrus via the pAF (posterior part), a result which is consistent with previous fMRI reports[40].

We observed that the pAF was involved in all forms of alexias, thus reinforcing the view that acquisition of literacy is associated with significant strengthening of this tract[3]. Our results suggest that the pAF may be dissociable into two functional strata, as fibers converging in the gyrus supramarginalis tended to be associated with Ph-A whereas fibers converging into the angular gyrus were rather associated with Ls-A—though this result was not strengthened by further pairwise disconnection comparisons probably due to a lack of statistical power (low sample size). This fits well with the suggested dissociated role of the gyrus supramarginalis and the angular gyrus in phonological and semantic processing, respectively[48–50].

The pivotal implication of the VWFA in word recognition is widely accepted[7,10] and damage to this area as well as deprivation of its connective inputs (i.e. fibers of the posterior ILF) may lead to Pu-A[22]. Our results not only confirm the role of the ILF but also highlight the possible implication of VOF disconnections in the event of Pu-A. The participation of the VOF in reading has been previously discussed[24,51], but to the best of our knowledge it is not experimentally evidenced. In this context, our results clearly contribute to the emerging view that this intra-occipital tract may integrate information in the service of reading by providing a neural route between the dorsal and the ventral visual stream[52].

No conclusive differences between Ph-A vs phonological paraphasia and Ls-A vs semantic paraphasia were observed. These findings are consistent with the primary systems hypothesis whereby reading is embedded within the neural and cognitive architecture of spoken language[53,54]. However, while a difference was found between Ph-A and anomia (related to marked disconnections of the AF in Ph-A), we failed to observe such a difference when anomia was compared to Ls-A. In the latter case, this is unsurprising given that the connectivity patterns related to each manifestation were indistinctively dominated by disconnections involving the ILF which is likely to participate to both the lexical-semantic aspect of reading (as shown in this study) and lexical retrieval (as reported in two recent neuromodulation or lesion mapping studies)[55,56].

From a clinical standpoint, our overall findings may help refine the way wide-awake neurosurgeries are performed in the vicinity of the VWFA. As fiber tracts are established to be less prone to neuroplastic compensation compared to cortical structures[57], unwavering attention must be paid to the sparing of white matter connectivity[38]. However, precise identification of a given tract with stimulation mapping is consubstantial with an advanced understanding of the processes conveyed by the tract in question. In this context, our results may serve as a basis for increasing the sensitivity of the intraoperative tasks used to monitor reading processes as well as for identifying their corresponding pathways; this may allow to decrease the likelihood of postoperative, irreversible alexia[58]. In the same way, these findings may help better predict the severity of reading disorders in multiple conditions wherein cerebral disconnections are the rule (e.g. stroke, traumatic brain injuries or brain neoplasms) and thus guide rehabilitation strategies to enhance cognitive recovery.

Several notes of caution must be mentioned not to overstate the current findings. First, as the electrostimulation dataset has been obtained in a particular clinical context, the behavioral paradigm used intraoperatively is necessarily reductionist. Although the stimulation trains were applied following a well-tried procedure allowing to control for false negative or positive outcomes[59], it remains that the number of stimulations per white matter locus was limited, thus implying that alexia classification was established on the basis of categorical diagnoses rather than dimensional diagnoses[21]. It should also be acknowledged that writing abilities were not assessed during the individual mappings, not allowing to rule out the possibility of concomitant agraphia in some cases. Second, we relied on population-averaged template of 1021 healthy subjects to derive streamline tractograms. This approach is certainly not optimal, but the use of diffusion data in patients with lesional infiltrations of the white matter is quite challenging and currently not validated. Third, even though electrostimulation mapping has a rather high spatial resolution compared to other lesional approaches, we cannot rule out the possibility that multiple white matter tracts could be inactivated during the same stimulation, especially in regions where tracts cross together or are in close proximity. It is a shortcoming that is hardly surmountable at this spatial scale, but not specific to the stimulation approach. This virtually implies that acquired alexia might arise, in certain cases, from multiple disconnections. Importantly, this might explain why it was more difficult to segregate disconnectivity patterns related to Pu-A in comparison to Ls-A. Fourth, disconnectivity matrices were generated using cortical parcels uniquely belonging to the left hemisphere, thus excluding the possibility to gauge the possible implication of the corpus callosum. We primarily opted for this methodological choice to reduce dimensionalities. However, it is worth recalling that interhemispheric tracts, especially the splenial portion of the corpus callosum, have potentially a role in reading by integrating the visual information from both occipital cortices[60]. Another limitation of this study is related to clinical and surgical constraints. As the spatial topography of each tumor could differ within the left hemisphere, the same stimulation sites were not tested strictly equally across all the subjects, potentially inducing a sampling bias. Last, tumors spreading in the vicinity of the VWFA are to the very least rare, explaining why only a limited number of white matter sites for alexia were available. Although the present dataset is unique, a larger one is needed to better dissociate the disconnectivity patterns related to Ls-A and Pu-A.

To conclude, our results shed light on the disconnective mechanisms underlying the main forms of alexia, with important clarifications regarding the contributions of the ILF, the VOF, the lsAF and the pAF. Taken together, these findings provide a strong anatomical background for current cognitive and neurocomputational models of reading processing.

## Methods

**Study design and patient selection.** Data used in this study were gained in a clinical context and the protocol described below reflects our standard clinical approach. All patients gave their informed consent. All methods of this study were performed in compliance with the tenets of the Declaration of Helsinki. Data and imaging were studied following anonymization in agreement with the Personal Data Protection Act and the code of conduct for responsible use of Human Tissue and Medical Research. Approval for the study and its methods was granted by the Institutional Review Board of Montpellier University Medical Center (No. 202000557).

We screened the intraoperative brain electrostimulation data obtained from consecutive awake surgeries performed in our institution from 2010 to 2020. To be included, patients had to fulfill the following criteria: (1) a diagnosis of diffuse low-grade glioma confirmed by histopathological and molecular analysis, (2) right-handedness as assessed by the Edinburgh inventory[61], (3) left-sided tumor location and (4) at least one axonal reading site identified by DES. According to these criteria, 25 patients were included (mean age: $39.4 \pm 11.4$ years, 6 were females and 19 were males). The mean presurgical tumor volume (based on fluid-attenuated inversion recovery sequences) was $36.6 \pm 23.5$ ml.

**Surgical procedure and intraoperative behavioral paradigms**. The surgical approach of cortico-subcortical DES mapping in patients undergoing asleep-awake-asleep craniotomy has been exhaustively reported elsewhere[29]. Briefly, following craniotomy, the cortical surface was exposed, and the edges of the tumor were visualized by intraoperative ultrasound and indicated with sterile letter tags. Once the patient was awake, electrical cortical mapping was performed with a bipolar electrode probe with a 5 mm inter-tip spacing (NIMBUS Stimulator, Newmedic, France), delivering a biphasic electric current (60 Hz, 1 ms pulse width, amplitude 1.75 to 3.50 mA). The amplitude was gradually increased until a positive response from the ventral premotor cortex was obtained (i.e. transient speech articulatory disturbances). This amplitude was not modified during the remainder of the intraoperative mapping (including both cortical and subcortical axonal mapping). In order to restrain the spread of the electric current to the adjacent neural structures, the duration of DES never exceeded 4 s.

After completion of the cortical mapping, subpial dissection and tumor removal was started, allowing an access to the white matter pathways. Subcortical axonal DES mapping was performed to achieve tumor resection according to individual functional boundaries. Importantly, a stimulation site was considered functional if DES elicited disturbances at least three times in a nonconsecutive manner[59]. Intraoperative assessment of language functions was achieved by a senior speech therapist who remained blinded to DES application.

First, intraoperative tasks included a motor task, a picture naming task[62], a semantic association task (Pyramids and Palm Trees test)[63] and a dual-task (picture naming plus movement)[64]. DES-induced language disturbances were recorded as follows: (1) inability to name visually presented pictures was accounted as anomia; (2) incorrect but semantically related picture naming (e.g. "cat" instead of "dog") was accounted as semantic paraphasia and (3) incorrect picture naming with phonemes substitution/rearrangement was accounted as phonological paraphasia (e.g. "tephelone" instead of "telephone"). Second, a reading aloud task derived from the ECLA 16 + test[65] was used. More specifically, 60 words (including 20 regular, 20 irregular and 20 pseudowords) controlled for psycholinguistic parameters, were consecutively and randomly presented on a laptop screen every 4 s.

A reading impairment was defined as an inability to read correctly a single word within the 4 s time delay. Reading impairments being reproducible at least three times during three nonconsecutive stimulations were considered as positive responses. More specifically, DES causing impairments in reading regular, irregular and pseudowords without additional spoken language disturbances were considered to induce pure alexia (i.e. Pu-A). By contrast, specific DES-related impairment of pseudowords reading was considered to generate phonological alexia (Ph-A). Last, DES uniquely associated with irregular words reading disorders was considered to generate lexical-semantic alexia (Ls-A). Positive sites were recorded with sterile tags.

To decrease the likelihood of false positive responses during the intraoperative testing, all patients were tested with the above-mentioned tasks before surgery to ensure their abilities to perform all tasks accurately during the intraoperative procedure. Failed items were excluded from the intraoperative materials. To facilitate the spatial positioning of stimulation sites, two series of photographs were taken during the surgery: one at the end of the cortical mapping and one at the end of the resection. An example of cortical and subcortical mapping is provided in Fig. 1.

**Imaging acquisition**. MRI acquisition (3 Teslas, Siemens Avento, Siemens Medical Systems) was performed for all patients the day before surgery and 3 months after surgery, including Fluid-attenuated inversion recovery (FLAIR) sequences and T1-weighted 3D Gadolinium-enhanced sequences. The tumor volume was measured with a dedicated software (Myrian, Intrasense, Montpellier, France). All 3-month postoperative MRI datasets were conformed to the standardized MNI space, using SPM12 implemented in MATLAB environment (Release 2018a, The MathWorks Inc., Natick, NA, USA); normalized MRIs were individually and systematically checked to ensure accuracy. Resection cavities were then drawn using the semi-automatic drawing tool in MRIcron software (NITRC, University of Massachusetts Medical School in Worcester, MA, USA).

**Spatial positioning of DES positive sites**. The MNI coordinates of each stimulation point were recorded using operative reports and intraoperative photographs. A similar method was previously described with a high level of inter-rater reliability[56]. To increase the accuracy of this positioning work, a 3D pial-mesh reconstruction of each normalized 3-month postoperative MRI was generated using BrainVISA/Anatomist Software (Version 5.0, CEA I2BM, CATI Neuroimaging, Inserm IFR49, and CNRS, France). This mesh allows a 3D navigation through anatomical structures and an automatic MNI coordinates-surface-matching of the 3D model (Fig. 1). Then, using the MarsBar Toolbox implemented in MATLAB, spherical volumes of interest (VOI) were generated for each stimulation point. To account for the different intensities of stimulation, VOI of 2.5 mm (equivalent to the spatial resolution of the bipolar probe), 3.0 mm and 3.5 mm were created.

**Disconnectivity analyses**. Individual VOIs of 2.5-mm were embedded into the population-averaged, HCP-1021-1mm diffusion dataset[66] in DSI studio Software (http://dsi-studio.labsolver.org). This dataset was obtained from a multishell diffusion scheme with b-values of 1000, 2000, and 3000 s/mm². The number of diffusion sampling directions were 90, 90, and 90, respectively. The diffusion data were then reconstructed in the MNI space using q-space diffeomorphic reconstruction[34,67]. the in-plane resolution was 1.25 mm. We further reconstructed fibers intersecting with each VOI (i.e. the "disconnected" fibers) using a deterministic tracking algorithm. A quantitative anisotropy threshold of 0.12 was selected to filter out background voxels. This threshold was generated automatically by DSI studio following Otsu's method and further slightly down modulated to reach a value associated with an acceptable rate of spurious fibers. The default step size (1.0 mm) parameters was applied and a 0.20 value of smoothing was chosen to avoid important constraints on current propagation direction. A total of 50000 seeds were placed. The maximum angular threshold was set at 50° (in line with previous works reporting angular threshold ranging from 40 to 80°)[34,68,69]. This threshold was initially adopted to minimize false positive tracts, the rate of which typically increases with larger values. To account for this potential selection bias, we repeated our streamline analysis with different tracking parameters (i.e. 40°, 50° and 60° angular threshold). The same results were obtained irrespective of the angular threshold (Supplementary Fig. 3).

The quantification of lesion-related interrupted streamlines is increasingly used to measure the "severity of structural disconnections" in tractographic lesion-based studies[70,71]. The reconstructed fibers were individually embedded in the Brainnetome atlas[33] (https://atlas.brainnetome.org/bnatlas, containing 123 labeled regions) to generate a 123 × 123 disconnectivity matrix related for each stimulation site (only left parcels were used). As a first step, disconnectivity matrices were summed by alexia subtype (grouped disconnection matrices are provided in Supplementary Data 1); this allowed us to obtain circular disconnectgrams which were useful for visual interpretation (RStudio version 1.3, R Foundation for Statistical Computing, Vienna, Austria; Chordiag 0.1.3 package).

In addition, serial pairwise comparisons were achieved between disconnectivity matrices related to Ph-A, Ls-A and Pu-A using serial two-tailed non-parametric Mann–Whitney tests. The goal here was to identify which pairwise white matter (dis)connections were specific to each alexia subtype. Statistical comparisons resulted in $n$ 123 × 123 matrix in which each matrix coordinate contained a Z-statistics further plotted in the form of heatmaps. Note that no adjustment for multiple correction was applied due to the exploratory nature of this complementary analysis.

**Tract-level analyses**. The total number of generated fibers using 2.5-mm VOIs were collected and projected onto the MNI space along with the HCP population-averaged atlas of the human connectome[34] using DSI studio software. The "recognize" function was used to filter fibers intersecting the ROIs and belonging to the main associative white matter pathways assumed to be involved in reading processes (i.e. the long segment [lsAF] and the posterior segment (pAF) of arcuate fasciculus [pAF], IFOF, ILF, and VOF). In this way, the number of "disconnected" fibers by tract could be estimated. Disconnected streamlines by tract were regrouped by alexia subtype (the number of disconnected streamlines by alexia subtype is provided in Supplementary Data 2) or language impairments (Ph-A vs Ls-A vs Pu-A vs phonological paraphasia vs semantic paraphasia vs anomia) and statistically compared using two-sided non-parametric ANOVAs (i.e. Kruskall–Wallis). Further pairwise comparisons were performed and corrected for multiple comparisons with the Dunn's test. Adjusted $p$ values < 0.05 were considered statistically significant. These analyses were repeated using VOIs of different sizes (3.0 and 3.5 mm) to account for the different intensities of stimulations (see Supplementary Fig. 2).

In addition, tracts generated with DSI studio were converted into 3D binarized maps and regrouped according to the clinical manifestations. Between-group voxel-based comparisons were then performed at the brainwide level using two-sided chi-square tests (Matlab in-house script). The subsequent statistical maps were statistically corrected using a FDR procedure with a $q = 0.05$.

**Statistics and reproducibility**. Non-parametric tests were systematically used. A significance level of 0.05 was used for all statistical tests. Serial pairwise comparisons between $n$ 123 × 123 matrices were performed with two-sided rank-based Mann–Whitney tests without adjustment for multiple comparisons due to the limited number of observations and the considerable number of comparisons. Tract analyses (based on streamline counts) were performed with two-sided Kruskall–Wallis tests and corrected for multiple comparisons with the Dunn's test. Tract analyses (3D binarized maps) were performed with two-sided chi-square tests and FDR-corrected with a $q = 0.05$ for multiple comparisons.

**Reporting summary**. Further information on research design is available in the Nature Research Reporting Summary linked to this article.

## Data availability

All Montreal Neurological Institute coordinates of positive stimulation points are provided in Supplementary Tables 1 and 2. Diffusion data are available on the HCP website (https://humanconnectome.org). Disconnection matrices by alexia subtype and individual streamline counts within the main white matter pathways are provided in

Supplementary Data 1 and 2, respectively. Other specific subsets of data that were used in the present study are available from the corresponding author upon request.

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

## Acknowledgements

S.N. was funded by a grant « Année-Recherche » from DGOS/ARS Occitanie.

## Author contributions

S.N.: investigation, software, visualization, methodology, writing—original draft. S.M.-G.: data curation, writing—review and editing. A.-L.L.: software, methodology, writing—review and editing. H.D.: project administration, supervision, data curation, validation, writing—review and editing. G.H.: project administration, conceptualization, software, methodology, supervision, writing—original draft.

## Competing interests

The authors declare no competing interests.
