## [Transparent Peer Review File · Communications Biology]

Reviewers' comments:

Reviewer #1 (Remarks to the Author):

In this article, Ng and collaborators perform in vivo direct electrostimulation (DES) in a series (n=25) of neurosurgical patients with low-grade glioma near the visual word form area in the left hemisphere to study and characterize the patterns of structural disconnectivity associated to different types of alexia (phonological, pure, and lexical-semantic).

Results are novel and provide key evidence on the neural substrates of acquired reading disorders. It has been a pleasure to read this article for its clarity and the presentation of such an interesting experiment. The figures are very informative and have a very careful aesthetic. Conclusions are well justified by the obtained results, and the potential weaknesses of the study are acknowledged in the discussion section. Statistical analyses are appropriate.

First, axonal DES caused reading difficulties in 25 patients. The coordinates of each stimulation site associated with each type of reading or speech disturbances are reported in Figure S1.

Second, structural disconnectivity for each alexia subtype was identified in the left hemisphere. To do that, they performed probabilistic tracking using as a seed a 2.5 mm VOI located in each individually responsive white matter site, in the HCP-1021 dataset, and the fibers intersecting each VOI were reconstructed. Pairwise disconnectivity matrices were created and disconnectograms were explored among alexia subtypes. This analysis revealed that phonological alexia was characterized by fronto-temporal and parieto-temporal disconnection mainly; lexico-semantic alexia was characterized by a different pattern of disconnection involving temporo-occipital, fronto-occipital and parieto-occipital disconnections; and pure alexia was associated by disconnection between temporo-occipital and intraoccipital disconnections (and less by temporo-parietal or parieto-occipital disconnections).

Third, major white matter tracts were identified among the tracked fibers and compared between each alexia type. The posterior segment of the arcuate fasciculus (AF) and the inferior longitudinal fasciculus (ILF) were associated with the three types of alexia without differences. The long segment of the AF was associated to alterations compatible with phonological alexia, whereas fibers of the vertical occipital fasciculus were more associated to pure alexia compared to lexico-semantic alexia. No conclusive differences were found between reading and language errors (e.g., phonological alexia and phonological paraphasia), likely due to the great overlap existing among language and reading systems.

Overall, the differential patterns found confirm the brain basis of the different cognitive strategies used to read (whole visual, lexico-semantic, phonological), which are related with each type of alexia when these areas are damaged. This is an indirect but elegant and smart form to study disconnection: axonal stimulation leads to specific reading errors associated to different types of known alexias, and fiber tracking from these sites performed in an independent sample may inform the circuitry implicated and the cortical areas that may suffer from indirect disconnection.

Line 222: there is a mistake, figure to be called is Figure 3a (corresponding to phonological alexia sites).

Reviewer #2 (Remarks to the Author):

In their article "White matter disconnectivity fingerprints causally linked to dissociated forms of alexia", the authors explore the white matter networks of reading through studying disconnection patterns.

This study used a combination of electrostimulation and tractography to investigate the white matter pathways associated with specific forms of alexia. In doing so, the authors exposed white matter pathways specific to each alexia form, improving our knowledge of the anatomical bases of the cognitive processes of reading. I think the article successfully advances our understanding of the brain networks of reading. However, I have a few comments and questions for the authors.

Main comments:

I558: "We further reconstructed fibers 559 intersecting with each VOI (i.e. the "disconnected" fibers) using a tracking threshold of 0.12, 560 an angular threshold of 50, a step size of 1.0 mm and a smoothing of 0.20." Unless I missed it, I do not see a justification for the thresholds and smoothing used here. More details about these choices or at least references showing that they are commonly used could prevent replication studies from blindly following these parameters.

I244 & I568: I am not sure to understand what is compared in the pair-wise comparison with the Mann-Whitney tests and whether a correction for multiple comparisons has been employed.

I581: If I understand correctly, the VOIs were used to count the disconnected fibres within a population-averaged diffusion atlas. Not many details about the tractography method are present in the paper. Is the tractography algorithm used deterministic? If yes, could you provide evidence that counting the number of fibres in deterministic tractography actually reflects the strength of the connectivity between brain regions?

Reviewer #3 (Remarks to the Author):

In their paper 'White matter disconnectivity fingerprints causally linked to dissociated forms of alexia', Ng and colleagues use intraoperative stimulation to selectively and temporarily disrupt activity along white matter tracts in order to identify the neuroanatomical substrates of reading. They identify distinct patterns of disconnection underlying three subtypes of alexia: pure alexia, phonological alexia, and lexical-semantic alexia. Pure alexia related to disruption of the inferior longitudinal fasciculus and vertical occipital fasciculus, phonological alexia was related to disruptions of the arcuate fasciculus that connects frontal and temporal regions, and lexical-semantic alexia was related to disruption of inferior longitudinal fasciculus and the interior fronto-occipital fasciculus.

Currently, the manuscript has a few technical limitations and is missing some key pieces of information that reduce my enthusiasm for its publication. However, the dataset is highly unique and specifically tailored to the question at hand; the authors use a non-invasive approach to causally determine the function of tracts within reading pathways by creating virtual transient lesions in white matter areas of interest. The paper is well-written and clear. I believe once the comments are addressed, the paper will have a measurable impact on the field.

1. How were stimulation sites determined, and were the same sites tested equally across all subjects? Were there stimulation sites that were not accessible intraoperatively in some subjects that may contribute to reduced disconnectivity estimates for these areas, and vice versa for more accessible areas? If so, discussion of the possible impact of this sampling bias should be included.
2. The description of findings under 'effects of direct electrostimulation on reading abilities' requires elaboration, as these results inform the alexia subtyping used in subsequent analyses. Was more than one subtype of alexia elicited in the same patient? If so, this should be communicated, possibly in the supplementary materials as a table indicating presence/absence of the alexia subtype elicited by DES for each subject.
3. The data presented in figure 3 are 'disconnectivity matrices summed by alexia subtype'. Given the

large sample size discrepancy between subtypes (n=4, n=7, and n=16), the relative differences in sample size appear to contribute most to this value. The median or mean disconnectivity between ROIs would convey more information regarding the relative amount of disconnection observed within each alexia subtype.

4. Regarding the visualization of Figure 3:

a. The chord diagrams as currently presented are not optimized for visually assessing the differences between alexia subtypes. This is because the location of the labels along the perimeter of the chord diagrams shifts based on the number of fibers disconnected in that lobe. Changing the representation of disconnections to a colormap, instead of line thickness, may be a solution. If the authors choose to keep the figures as-is, a statement in the figure description stating that shifts in label location occur should be included.

b. The location of the axonal DES sites could be presented in an additional view (i.e. inferior and/or posterior) to see where in the white matter the stimulation occurred; it is hard to determine WM depth/location with only a lateral view.

5. The name of the ROI of the orbital gyrus is missing from the following sentence: "Most of the frontal lobe disconnections were related to the orbital gyrus."

6. The pairwise disconnectivity comparisons between alexia subtypes are not corrected for multiple comparisons, but the authors have performed a test at each element of the upper triangular of the disconnectivity matrix, and therefore these results should be corrected, or it should at least be noted that the results do not survive correction. It is also unclear why the analyses were presented as one-tailed tests (taking $Z > 1.96$ and not $|Z| > 1.96$) when the authors are looking for differences between groups in either direction. The small sample size is also a concern in the analyses (but is understandable with a rare dataset such as this), as is the potential non-independence of data points in the case that a subject is included in more than one subtype. Perhaps the authors could display a histogram or violin plot of the disconnection scores for the most significantly different pairings (i.e., those plotted in Figure 4d/mentioned by name in the results) to confirm that the differences in disconnectivity are not related to outliers in the smaller sample size.

7. The font size in the chord diagrams is not easily read, especially in Figure 4d. Several ROIs are also plotted twice around the chord diagram; this should be explicitly stated in the figure caption to avoid confusing readers who have been oriented to chord diagrams in Figure 3. An alternative could be plotting the results separately for the three comparisons.

8. The tract-level analyses do not confirm, in the sense of validation, as much as contextualize the pairwise disconnectivity findings within canonical reading-related white matter tracts. The statement "To confirm the patterns of disconnectivity describe above..." should be modified accordingly to not confuse the reader.

9. "There were no statistical differences in the number of disconnected fibers between stimulations soliciting Ph-A and phonological paraphasias." The figures/results of this analysis should be referenced.

10. In Figure 5, only the extreme values (whiskers) of the box plots are plotted; a violin plot or other figure that displays individual data points would add clarity to this figure.

11. The unique relationship between disruption to the pAF and ph-A is not well-supported by the data (seemingly only in Figure 3a) -- much more evidence points to the relationship between the IsAF and ph-A. However, the former result is more heavily represented in the discussion.

Emily Olafson

Manuscript number: COMMSBIO-21-2480

Title: White matter disconnectivity fingerprints causally linked to dissociated forms of alexia

Reviewer #1 (Remarks to the Author):

In this article, Ng and collaborators perform in vivo direct electrostimulation (DES) in a series (n=25) of neurosurgical patients with low-grade glioma near the visual word form area in the left hemisphere to study and characterize the patterns of structural disconnectivity associated to different types of alexia (phonological, pure, and lexical-semantic).

Results are novel and provide key evidence on the neural substrates of acquired reading disorders. It has been a pleasure to read this article for its clarity and the presentation of such an interesting experiment. The figures are very informative and have a very careful aesthetic. Conclusions are well justified by the obtained results, and the potential weaknesses of the study are acknowledged in the discussion section. Statistical analyses are appropriate.

First, axonal DES caused reading difficulties in 25 patients. The coordinates of each stimulation site associated with each type of reading or speech disturbances are reported in Figure S1.

Second, structural disconnectivity for each alexia subtype was identified in the left hemisphere. To do that, they performed probabilistic tracking using as a seed a 2.5 mm VOI located in each individually responsive white matter site, in the HCP-1021 dataset, and the fibers intersecting each VOI were reconstructed. Pairwise disconnectivity matrices were created and disconnectograms were explored among alexia subtypes. This analysis revealed that phonological alexia was characterized by fronto-temporal and parieto-temporal disconnection mainly; lexico-semantic alexia was characterized by a different pattern of disconnection involving temporo-occipital, fronto-occipital and parieto-occipital disconnections; and pure alexia was associated by disconnection between temporo-occipital and intraoccipital disconnections (and less by temporo-parietal or parieto-occipital disconnections).

Third, major white matter tracts were identified among the tracked fibers and compared between each alexia type. The posterior segment of the arcuate fasciculus (AF) and the inferior longitudinal fasciculus (ILF) were associated with the three types of alexia without differences. The long segment of the AF was associated to alterations compatible with phonological alexia, whereas fibers of the vertical occipital fasciculus were more associated to pure alexia compared to lexico-semantic alexia. No conclusive differences were found between reading and language errors (e.g., phonological alexia and phonological paraphasia), likely due to the great overlap existing among language and reading systems.

Overall, the differential patterns found confirm the brain basis of the different cognitive strategies used to read (whole visual, lexico-semantic, phonological), which are related with each type of alexia when these areas are damaged. This is an indirect but elegant and smart form to study disconnection: axonal stimulation leads to specific reading errors associated to different types of known alexias, and fiber tracking from these sites performed in an independent sample may inform the circuitry implicated and the cortical areas that may suffer from indirect disconnection.

Line 222: there is a mistake, figure to be called is Figure 3a (corresponding to phonological alexia sites).

The authors: Thank you very much for your encouraging comments and your careful reading. The mistake pointed by the reviewer has been corrected:

Changes to the result section:

“The disconnectivity pattern of Ph-A was characterized by fronto-temporal (involving the inferior frontal gyrus [IFG] and the middle frontal gyrus [MFG] on the one hand, and the middle temporal gyrus [MTG] and the inferior temporal gyrus [ITG] on the other hand) and parieto-temporal disconnections (involving pairwise disconnections between the inferior parietal lobule [IPL] and both the MTG and ITG) (Figure 3a).” line 201

Reviewer #2 (Remarks to the Author):

In their article "White matter disconnectivity fingerprints causally linked to dissociated forms of alexia", the authors explore the white matter networks of reading through studying disconnection patterns.

This study used a combination of electrostimulation and tractography to investigate the white matter pathways associated with specific forms of alexia. In doing so, the authors exposed white matter pathways specific to each alexia form, improving our knowledge of the anatomical bases of the cognitive processes of reading. I think the article successfully advances our understanding of the brain networks of reading. However, I have a few comments and questions for the authors.

The authors: We thank the reviewer 2 for her/his positive appreciation of our manuscript. Please see our point-by-point response to the reviewer's comments/issues.

1. Main comments:

I558: "We further reconstructed fibers 559 intersecting with each VOI (i.e. the "disconnected" fibers) using a tracking threshold of 0.12, 560 an angular threshold of 50, a step size of 1.0 mm and a smoothing of 0.20." Unless I missed it, I do not see a justification for the thresholds and smoothing used here. More details about these choices or at least references showing that they are commonly used could prevent replication studies from blindly following these parameters.

The authors: We acknowledge that some methodological details and justifications were lacking in the Methods section of the manuscript. First, the "tracking threshold" (used value 0.12), which corresponds to the quantitative anisotropy (QA) threshold, was used as a mask to filter out background voxels. This threshold was generated automatically by DSI studio software using the Otsu's method and further slightly down modulated to reach a value associated with an acceptable rate of spurious fibers. Regarding the smoothing, a low value of 0.2 was chosen to avoid important constraints on current propagation direction. The maximum angular threshold was set at 50° (in line with previous works reporting angular threshold ranging from 40 to 80°) [Yeh et al, PLoS One 2013, Girard et al, Neuroimaging 2018, Wu et al, Front Neuroanatomy 2016, Yeh et al Neuroimage 2018]. This threshold was initially adopted to minimize false positive tracts, the rate of which typically increases with larger values. To account for this potential selection bias (higher angular thresholds might allow the tracking of fibers with more abrupt turning), we repeated our streamline analysis with different tracking parameters (i.e. angular threshold 40°, 50° and 60°). We found no significant difference in the distribution of estimated streamline disconnections (see Supplemental Figure 3).

Changes to the Methods section:

"We further reconstructed fibers intersecting with each VOI (i.e. the "disconnected" fibers) using a deterministic tracking algorithm. A quantitative anisotropy threshold of 0.12 was selected to filter out background voxels. This threshold was generated automatically by DSI studio following Otsu's method and further slightly down modulated to reach a value associated with an acceptable rate of spurious fibers. The default step size (1.0 mm) parameters was applied and a 0.20 value of smoothing was chosen to avoid important constraints on current propagation direction. A total of 50000 seeds were placed. The maximum angular threshold was set at 50° (in line with previous works reporting angular threshold ranging from 40 to 80°). [Yeh et al, PLoS One 2013, Wu et al, Front Neuroanatomy 2016, Yeh et al Neuroimage 2018]. This threshold was initially adopted to minimize false positive tracts, the rate of which typically increases with larger values. To account for this potential selection bias, we repeated our streamline analysis with different tracking parameters (i.e. 40°, 50° and 60° angular threshold). The same results were obtained irrespective of the angular threshold (Supplemental Figure 3)." Line 584

Addition to the Supplementary data:

Supplementary Figure 3. Tract-level analyses. Distribution of “disconnected” streamlines by angular threshold.

Main white matter tracts (lsAF: long segment of the arcuate fasciculus, pAF: posterior segment of the arcuate fasciculus, ILF: inferior longitudinal fasciculus, IFOF: inferior fronto-occipital fasciculus, VOF: vertical occipital fasciculus) according to the HCP-1065 white matter tractography atlas and number of disconnected fibers in each form of alexia (Ph-A: phonological alexia, Ls-A: lexical-semantic alexia, Pu-A: pure alexia) and speech disorders (Ph-P: phonological paraphasia, Se-P: semantic paraphasia, An: anomia). The total number of generated streamlines using a maximum angular threshold of 40° (left), 50° (middle) and 60°(right) were collected and projected into the MNI space along with the HCP population-averaged atlas of the human connectome using DSI studio software.

Individual values are presented as dots. Horizontal lines represent the median values.

Kruskall-Wallis tests were used to performed group comparisons. P-values were adjusted with the Dunn’s test for multiple comparisons. * $p < 0.05$; ** $p < 0.01$; *** $p < 0.001$

2. **I244 & I568: I am not sure to understand what is compared in the pair-wise comparison with the Mann-Whitney tests and whether a correction for multiple comparisons has been employed.**

The authors: We apologize if the manuscript lacks clarity about this sub-analysis.

For each patient, all parcel-to-parcel disconnections were converted into disconnectivity matrices (123*123, according to the number of parcels within the left hemisphere, as defined by the Brainnetome atlas). Then, two-tailed Mann-Whitney tests were performed to compare the values contained in each coordinate of the 123*123 matrices between Ph-A vs Ls-A, Ph-A vs Pu-A and Ls-A vs Pu-A.

The results of this analysis were not corrected for multiple comparisons due to both the limited amount of data for each alexia subtype, especially for phonological alexia ($n = 4$) and, conjointly, the considerable number of statistical comparisons ($n = 7503$ by comparison).

The smallest p -values that can be obtained from non-parametric comparisons (with two-tailed Mann-Whitney tests) given the sample sizes will not theoretically survive the corrections for multiple comparisons, whatever are the ranks of observations. For instance, the smallest theoretical p -value that can be obtained by comparing 4 vs 16 observations [Ph-A vs Ls-A] is $P=0.002$, while the smallest theoretical p -value that can be obtained by comparing 4 vs 7 observations [Ph-A vs Pu-A] is $P=0.008$. Accordingly, we clearly indicated in the new version of the manuscript that these results should be interpreted with caution. Note that these results were not over-interpreted and considered as additional in the previous version of the work.

Please see the following changes:

Changes to the Results section:

“These analyses resulted in n 123x123 matrices in which each matrix coordinate held a Z statistics. These matrices were further plotted in the form of heatmaps (Figure 4a-c). Disconnections that were considered different from one alexia subtype to another (i.e. $Z > 1.96$, $p < 0.05$ uncorrected) are displayed in Figure 4a-c. Note that the following comparisons have to be interpreted with caution, since no adjustments for multiple comparisons were performed due to the limited amount of data for each alexia subtype, especially for phonological alexia ($n = 4$) and, conjointly, the considerable number of statistical comparisons. Overall, pairwise connections between MFG and ITG (Z ranged from 2.37 to 3.02; uncorrected p -value ranged from 0.02 to 0.002) and between IFG and ITG (Z ranged from 1.98 to 2.65; uncorrected p -value ranged from 0.04 to 0.008) were more frequently disrupted in the event of Ph-A compared with Ls-A and Pu-A. Interestingly, impaired connections between the IFG and the anterior-lateral part of the VWFA (ITG_L_7_7) and between the MFG and the anterior-lateral part of the VWFA were specifically found in the event of Ph-A. In addition, impaired ITG-angular connections had a higher propensity to be disrupted in Ls-A than in Pu-A ($Z = 1.97$; $p = 0.04$ uncorrected).” Line 223

Changes to Figure 4 caption:

“^aNo corrections for multiple comparisons were applied”

Changes to the Methods section:

“Statistical comparisons resulted in n 123x123 matrix in which each matrix coordinate contained a Z-statistics further plotted in the form of heatmaps. Note that no adjustment for multiple correction was applied due to the exploratory nature of this complementary analysis.” Line 612

Addition to the Statistics and Reproducibility section:

“Serial pairwise comparisons between n 123x123 matrices were performed with two-tailed rank-based Mann-Whitney tests without adjustment for multiple comparisons due to the limited number of observations and the considerable number of comparisons.” Line 645

3. I581: If I understand correctly, the VOIs were used to count the disconnected fibres within a population-averaged diffusion atlas. Not many details about the tractography method are present in the paper. Is the tractography algorithm used deterministic? If yes, could you provide evidence that counting the number of fibres in deterministic tractography actually reflects the strength of the connectivity between brain regions?

The authors: Individual VOIs were embedded in a population-averaged diffusion atlas (i.e. the averaged diffusion data of the 1065 HCP participants) to generate “disconnected” streamlines. A deterministic tracking algorithm was employed. This method was preferred over a probabilistic fiber tracking because it was previously demonstrated to achieve a much higher rate of valid connections, although it appears to be less effective to track short-range connections (Maier-Hein et al, Nature Communications 2016). The number of streamlines reconstructed between two regions provides a quantitative measure of structural (dis)connectivity between these two regions, but it does not estimate the “strength of connectivity” *per se*. To the authors’ knowledge, no index derived from tractography truly reflect the “strength of connectivity” in a physiological or anatomical context, [Jones et al, Neuroimage 2013]. Having said that, the quantification of lesion-related “interrupted” streamlines is increasingly used to measure the severity of structural disconnections in tractographic lesion-based studies (Jones et al, Neuroimage 2013, Griffis et al, Cell report 2019, Griffis et al, Neuroimage Clinical 2021).

Changes to the Methods section:

“The quantification of lesion-related interrupted streamlines is increasingly used to measure the severity of structural disconnections in tractographic lesion-based studies (Griffis et al, Cell report 2019, Griffis et al, Neuroimage clinical 2021).” Lines 599

Reviewer #3 (Remarks to the Author):

In their paper ‘White matter disconnectivity fingerprints causally linked to dissociated forms of alexia’, Ng and colleagues use intraoperative stimulation to selectively and temporarily disrupt activity along white matter tracts in order to identify the neuroanatomical substrates of reading. They identify distinct patterns of disconnection underlying three subtypes of alexia: pure alexia, phonological alexia, and lexical-semantic alexia. Pure alexia related to disruption of the inferior longitudinal fasciculus and vertical occipital fasciculus, phonological alexia was related to disruptions of the arcuate fasciculus that connects frontal and temporal regions, and lexical-semantic alexia was related to disruption of inferior longitudinal fasciculus and the interior fronto-occipital fasciculus.

Currently, the manuscript has a few technical limitations and is missing some key pieces of information that reduce my enthusiasm for its publication. However, the dataset is highly unique and specifically tailored to the question at hand; the authors use a non-invasive approach to causally determine the function of tracts within reading pathways by creating virtual transient lesions in white matter areas of interest. The paper is well-written and clear. I believe once the comments are addressed, the paper will have a measurable impact on the field.

The authors: We thank the reviewer 3 for her detailed appreciation of the manuscript and for her constructive methodological suggestions. We hope that the following clarifications and add-ons will mitigate the issues pointed by the reviewer.

1. How were stimulation sites determined, and were the same sites tested equally across all subjects? Were there stimulation sites that were not accessible intraoperatively in some subjects that may contribute to reduced disconnectivity estimates for these areas, and vice versa for more accessible areas? If so, discussion of the possible impact of this sampling bias should be included.

The authors: We thank the reviewer for her valuable comment.

Due to clinical constraints, it was not possible to test equally all stimulation sites across all subjects. Indeed, tumors were not equally distributed among the left hemisphere (as illustrated in Figure 2). Resections were centered on the location of the tumor and white matter stimulations were applied at the edge of the resection cavity to guide the neurosurgical procedure (positive stimulations lead the surgeon to stop the resection, to prevent further neurological deterioration).

To account for this sampling bias, we mentioned this limitation in the discussion section – as suggested.

Changes to the Discussion section:

“Another limitation of this study is related to clinical and surgical constraints. As the spatial topography of each tumor could differ within the left hemisphere, the same stimulation sites were not tested strictly equally across all patients, potentially inducing a sampling bias.” Line 475

- 2. The description of findings under ‘effects of direct electrostimulation on reading abilities’ requires elaboration, as these results inform the alexia subtyping used in subsequent analyses. Was more than one subtype of alexia elicited in the same patient? If so, this should be communicated, possibly in the supplementary materials as a table indicating presence/absence of the alexia subtype elicited by DES for each subject.**

The authors: We thank the reviewer for this suggestion.

A reading impairment was defined as an inability to read correctly a single word within a 4-sec time frame. Each reading impairment and the type of words that was not correctly read (regular, irregular, pseudo-words) was recorded to define the subtype of alexia induced by DES. The effects of DES on reading abilities was assessed by a senior speech therapist who remained blinded to DES application.

Overall, 27 positive functional sites were recorded in 25 patients. In only two patients, two distinct sites of alexia were identified, including one patient with two alexia subtypes. This information is now provided in the Supplementary Table 1 and was also added to the result section of the main manuscript.

Changes to the Results section:

“Biphasic axonal DES (60 Hz, 1 ms pulse width, mean intensity of 2.52 ± 0.58 mA ranging from 1.75 to 3.50 mA) caused transient difficulty in reading words in 25 patients. A reading impairment was defined as an inability to read correctly a single word within a 4-sec time frame. Intraoperative findings were recorded and interpreted by a senior speech-language pathologist who remained blinded to DES application. Overall, 72 white matter sites were identified (27 eliciting reading disturbances and 45 eliciting language disturbances). Note that two distinct sites of alexia were identified in two patients.” Line 128

Changes to the Supplementary table 1:

^a Ls-A-S5 and Pu-A-S4 were recorded in the same patient

^b Pu-A-S2 and Pu-A-S3 were recorded in the same patient

- 3. The data presented in figure 3 are ‘disconnectivity matrices summed by alexia subtype’. Given the large sample size discrepancy between subtypes (n=4, n=7, and n=16), the relative differences in sample size appear to contribute most to this value. The median or mean disconnectivity between ROIs would convey more information regarding the relative amount of disconnection observed within each alexia subtype.**

The authors: We are grateful to the Reviewer for her important suggestion.

We now applied a similar measurement scale to the three diagrams. In other words, the same average amount of disconnections by patient is now illustrated by the same line thickness within the three chord diagrams to allow easier and more comprehensive comparisons between alexia subtypes.

Of note, within a single diagram (at a given measurement scale), the distribution of the lines and the thickness of the lines are strictly proportional to the amount of disconnections and were not different whether the sum of disconnections or the average amount of disconnections were selected to generate the diagram (this is the reason why no changes were visible for the chord diagram “a” [Ph-A]).

4. Regarding the visualization of Figure 3:

a. The chord diagrams as currently presented are not optimized for visually assessing the differences between alexia subtypes. This is because the location of the labels along the perimeter of the chord diagrams shifts based on the number of fibers disconnected in that lobe. Changing the representation of disconnections to a colormap, instead of line thickness, may be a solution. If the authors choose to keep the figures as-is, a statement in the figure description stating that shifts in label location occur should be included.

The authors: In accordance with our previous response, we have now aligned and standardized the thickness of the lines on the average amount of disconnection per patient in all diagrams. For this reason, and with respect to the comment of the reviewer, we think that Figure 3 is now better optimized to allow appropriate visual comparison between alexia subtypes. As suggested by the reviewer, we added a statement about the shifts of the labels in the description of the figure.

Changes to the Figure 3 caption:

“The thickness of the chords is proportional to the average number of pairwise disconnections between brain areas. The location of the labels along the perimeter of the diagram is shifted from one diagram to another, depending on the average number of disconnections in a given diagram.” Line 185

b. The location of the axonal DES sites could be presented in an additional view (i.e. inferior and/or posterior) to see where in the white matter the stimulation occurred; it is hard to determine WM depth/location with only a lateral view.

The authors: We thank the reviewer for her suggestion and we agree. We added inferior and posterior views in the Figure 3.

Changes to the Figure 3:
see above, response R3Q3.

5. **The name of the ROI of the orbital gyrus is missing from the following sentence: “Most of the frontal lobe disconnections were related to the orbital gyrus.”**

The authors: We added the appropriate ROI reference in the manuscript.

Changes to the Results section:
“Most of the frontal lobe disconnections were related to the orbital gyrus (OrG_L_6_4).” Line 210

6. **The pairwise disconnectivity comparisons between alexia subtypes are not corrected for multiple comparisons, but the authors have performed a test at each element of the upper triangular of the disconnectivity matrix, and therefore these results should be corrected, or it should at least be noted that the results do not survive correction. It is also unclear why the analyses were presented as one-tailed tests (taking $Z > 1.96$ and not $|Z| > 1.96$) when the authors are looking for differences between groups in either direction. The small sample size is also a concern in the analyses (but is understandable with a rare dataset such as this), as is the potential non-independence of data points in the case that a subject is included in more than one subtype.**

The authors: As suggested by the reviewer, we added a statement both in the manuscript and the figure to inform the reader that pairwise disconnectivity comparisons were not corrected for multiple comparisons.

Concerning correction for multiple comparisons

The results of this analysis were not corrected for multiple comparisons due to both the limited amount of data for each alexia subtype, especially for phonological alexia ($n = 4$) and, conjointly, the considerable number of statistical comparisons ($n = 7503$ by comparison).

The smallest p -values that can be obtained from non-parametric comparisons (with two-tailed Mann-Whitney tests) given the sample sizes will not theoretically survive the corrections for multiple comparisons, whatever are the ranks of observations. For instance, the smallest theoretical p -value that can be obtained by comparing 4 vs 16 observations [Ph-A vs Ls-A] is $P=0.002$, while the smallest theoretical p -value that can be obtained by comparing 4 vs 7 observations [Ph-A vs Pu-A] is $P=0.008$. Accordingly, we clearly indicated in the new version of the manuscript that these results should be interpreted with caution. Note that these results were not over-interpreted and considered as additional in the previous version of the work.

Concerning one-tailed/two-tailed tests

Our apologies for the lack of clarity, but we confirm that all analyses were two-tailed. Z-value were presented as absolute values only for the need of graphical presentation (this was changed in the revised Figure).

Concerning the potential non-independence of data points

Due to the fact that only one subject presented with two alexia subtypes (Pu-A and Ls-A, see R3Q2) at two different stimulation sites, we consider this only as a minor issue here.

Changes to the Results section:

“we conducted pairwise comparisons between disconnectivity matrices related to Ph-A, Ls-A and Pu-A using **two-tailed** non-parametric Mann-Whitney tests” Line 222

“These analyses resulted in n 123x123 matrices in which each matrix coordinate held a Z statistics. These matrices were further plotted in the form of heatmaps (Figure 4a-c). Disconnections that were **considered different from one alexia subtype to another (i.e. $Z > 1.96$, $p < 0.05$ uncorrected)** are displayed in Figure 4a-c. Note that the following comparisons have to be interpreted with caution, since no adjustments for multiple comparisons were performed due to the limited amount of data for each alexia subtype, especially for phonological alexia ($n = 4$) and, conjointly, the considerable number of statistical comparisons. Overall, pairwise connections between MFG and ITG (Z ranged from 2.37 to 3.02; **uncorrected** p -value ranged from 0.02 to 0.002) and between IFG and ITG (Z ranged from 1.98 to 2.65; **uncorrected** p -value ranged from 0.04 to 0.008) were more frequently disrupted in the event of Ph-A compared with Ls-A and Pu-A. Interestingly, impaired connections between the IFG and the anterior-lateral part of the VWFA (ITG_L_7_7) and between the MFG and the anterior-lateral part of the VWFA were specifically found in the event of Ph-A. In addition, impaired ITG-angular connections had a higher propensity to be disrupted in Ls-A than in Pu-A ($Z = 1.97$; $p = 0.04$ **uncorrected**).” Line 223

Changes to Figure 4 caption:

^aNo corrections for multiple comparisons were applied”

Changes to the Methods section:

“In addition, serial pairwise comparisons were achieved between disconnectivity matrices related to Ph-A, Ls-A and Pu-A using serial **two-tailed** non-parametric Mann-Whitney tests.” Line 609

“Statistical comparisons resulted in n 123x123 matrix in which each matrix coordinate contained a Z-statistics further plotted in the form of heatmaps. Note that **no adjustment for multiple correction** was applied due to the **exploratory nature of this complementary analysis.**” Line 612

Addition to the Statistics and Reproducibility section:

“Serial pairwise comparisons between n 123x123 matrices were performed with two-tailed rank-based Mann-Whitney tests without adjustment for multiple comparisons due to the limited number of observations and the considerable number of comparisons.” Line 645

- Perhaps the authors could display a histogram or violin plot of the disconnection scores for the most significantly different pairings (i.e., those plotted in Figure 4d/mentioned by name in the results) to confirm that the differences in disconnectivity are not related to outliers in the smaller sample size.

The authors: We added violin plots to the Figure 4, in order to illustrate the number of disconnections between the most significantly different pairings. We also decided to remove the chord diagram (Figure 4d) and to add graphical representations of the parcels showing the highest $|Z|$ values.

Changes to the Figure 4:

8. The font size in the chord diagrams is not easily read, especially in Figure 4d. Several ROIs are also plotted twice around the chord diagram; this should be explicitly stated in the figure caption to avoid confusing readers who have been oriented to chord diagrams in Figure 3. An alternative could be plotting the results separately for the three comparisons.

The authors: To avoid confusion, we decided to remove the chord diagram (Figure 4d) and to add graphical representation of the parcels showing the highest $|Z|$ values.

Changes to the Figure 4:

See above, R3Q7.

9. The tract-level analyses do not confirm, in the sense of validation, as much as contextualize the pairwise disconnectivity findings within canonical reading-related white matter tracts. The statement “To confirm the patterns of disconnectivity describe above...” should be modified accordingly to not confuse the reader.

The authors: We thank the reviewer for her suggestion, which has been taken into consideration in the revised manuscript.

Changes to the results section:

Tract-level analyses

“To better delineate the patterns of disconnectivity described above” Line 283

10. “There were no statistical differences in the number of disconnected fibers between stimulations soliciting Ph-A and phonological paraphasias.” The figures/results of this analysis should be referenced.

The authors: We rephrased this sentence and added a reference to the Supplementary figure 2.

Changes to the Results section:

“There were no statistical differences in the number of “disconnected” streamlines within the main white matter tracts between stimulations eliciting Ph-A and phonological paraphasias (Supplementary figure 2).” Line 311

11. In Figure 5, only the extreme values (whiskers) of the box plots are plotted; a violin plot or other figure that displays individual data points would add clarity to this figure.

The authors: We added violin plots (instead of box plots) with individual data points to the Figure 5. Other figures included in the Supplemental data are now also presented as violin plots.

Changes to the Figure 5:

12. The unique relationship between disruption to the pAF and ph-A is not well-supported by the data (seemingly only in Figure 3a) -- much more evidence points to the relationship between the IsAF and ph-A. However, the former result is more heavily represented in the discussion. Emily Olafson

The authors: We toned down our interpretation regarding the role of the pAF in phonological alexia.

Changes to the Discussion section:

“Connections projecting into the gyrus supramarginalis via the pAF were associated with Ph-A, although such connectivity profile was not confirmed by further pairwise disconnection comparisons.” (lines 359)

“We observed that the pAF was involved in all forms of alexias, thus reinforcing the view that acquisition of literacy is associated with significant strengthening of this tract. Our results suggest that the pAF may be dissociable into two functional strata, as fibers converging in the gyrus supramarginalis tended to be associated with Ph-A whereas fibers converging into the angular gyrus were rather associated with Ls-A – though this result was not strengthened by further pairwise disconnection comparisons probably due to a lack of statistical power (low sample size)” (line 392)

REVIEWERS' COMMENTS:

Reviewer #1 (Remarks to the Author):

I have nothing more to contribute, just congratulate the authors for the great work.

Reviewer #2 (Remarks to the Author):

The authors answered my questions and concerns. I think the article is ready to be accepted.

Reviewer #3 (Remarks to the Author):

The authors have thoroughly addressed my comments and questions, and I am very satisfied with the revisions made to the manuscript.